# Reactive Oxygen Species and Long Non-Coding RNAs, an Unexpected Crossroad in Cancer Cells

**DOI:** 10.3390/ijms231710133

**Published:** 2022-09-04

**Authors:** Teodor Paul Kacso, Renata Zahu, Alexandru Tirpe, Elina Valeria Paslari, Andreea Nuțu, Ioana Berindan-Neagoe

**Affiliations:** 1Faculty of Medicine, “Iuliu Hațieganu” University of Medicine and Pharmacy, 8 Victor Babes Street, 400012 Cluj-Napoca, Romania; 2Department of Oncology and Radiotherapy, “Iuliu Hațieganu” University of Medicine and Pharmacy, 400347 Cluj-Napoca, Romania; 3Amethyst Radiotherapy Center Cluj, 407280 Florești, Romania; 4Research Center for Functional Genomics, Biomedicine and Translational Medicine, “Iuliu Hațieganu” University of Medicine and Pharmacy, 23 Marinescu Street, 400337 Cluj-Napoca, Romania; 5The Oncology Institute “Prof. Dr. Ion Chiricuta”, 34-36 Republicii Street, 400015 Cluj-Napoca, Romania

**Keywords:** lncRNAs, ROS, antioxidant response, cancer metabolism, chemoresistance

## Abstract

Long non-coding RNAs (lncRNA) have recently been identified as key regulators of oxidative stress in several malignancies. The level of reactive oxygen species (ROS) must be constantly regulated to maintain cancer cell proliferation and chemoresistance and to prevent apoptosis. This review will discuss how lncRNAs alter the ROS level in cancer cells. We will first describe the role of lncRNAs in the nuclear factor like 2 (Nrf-2) coordinated antioxidant response of cancer cells. Secondly, we show how lncRNAs can promote the Warburg effect in cancer cells, thus shifting the cancer cell’s “building blocks” towards molecules important in oxidative stress regulation. Lastly, we explain the role that lncRNAs play in ROS-induced cancer cell apoptosis and proliferation.

## 1. Introduction

Cancer remains the second leading cause of mortality worldwide, lagging only to cardiovascular disease [1]. Globally, an estimated 19.3 million new cancer cases (18.1 million excluding nonmelanoma skin cancer) and almost 10.0 million cancer deaths (9.9 million excluding nonmelanoma skin cancer) occurred in 2020 [2]. Moreover, the median age of cancer diagnosis is on a decreasing trend [3] and estimates suggest that this century could witness cancer surpassing cardiovascular disease as the number one cause of premature death in most countries [4]. However, constant research in the not-so-long-ago cryptic mechanisms of cancer cells has led to significant progress over the last decade, which can be readily seen in a 50% decrease (from 186.9 to 86.3) in the age-standardized mortality rate [2,5]. With that in mind, this review aims to coalesce the current knowledge and highlight the latest advancements regarding two important molecules in cancer cells: long non-coding RNAs (lncRNAs) and reactive oxygen species (ROS).

ROS are oxygen containing molecules characterized by their high reactivity, playing the role of oxidant or reductor in a wide range of redox reactions [6]. Though a heterogenous group, a few representatives are of significant importance in cancer: superoxide anion (•O_2_^−^), hydrogen peroxide (H_2_O_2_) and hydroxyl radical (•OH) [7]. These molecules can easily derive from one another through commonly occurring reactions throughout the cell. For example, •O_2_^−^ can be oxidized to H_2_O_2_ by superoxide dismutase (SOD) and H_2_O_2_ can be reduced by Fenton reaction to form •OH—a highly reactive and DNA damaging ROS [8].
$$O_{2}\;{+\;e}^{-}\to •O_{2}{}^{-}$$
$${•O}_{2}{}^{-}\overset{{}^{SOD}}{\to }H_{2}O_{2}$$
$$H_{2}O_{2}\;{+\mathrm{Fe}}^{2+}\to •{\mathrm{HO}+\mathrm{HO}}^{-}{+\mathrm{Fe}}^{3+}$$

In physiological conditions, ROS are not merely a byproduct of the cell’s redox homeostasis but do in fact carry out important functions [9]. For example, through the formation of disulfide bonds between thiol groups in adjacent cysteine residues, ROS can alter the function of proteins in key signaling pathways such as such as the PI3K/Akt, NF-κB, and p53 pathways [10,11,12]. However, due to their high reactivity, ROS need to be kept in check by well-orchestrated antioxidant machinery. An imbalance in oxidative stress regulation is a core pathophysiological process in multiple pathologies such as cardiovascular disease [13], diabetes [14], inflammatory disorders [15], and cancer.

Elevated levels of ROS are considered essential in promoting cancer cell growth and proliferation by augmenting oncogenes expression or inhibiting tumor suppressor genes, either by direct DNA damage or by post-translational modifications in key signaling proteins or enzymes [7,16,17,18]. Figure 1 reviews the most important ROS-coordinated pathways in the hallmarks of cancer [19,20,21]. On the other hand, high levels of ROS can overwhelm the antioxidant machinery of cancer cells and lead the cell to apoptosis [22]. As such, the ROS balancing mechanism is of paramount importance in cancer [23]. For example higher activity of antioxidant enzymes such as SOD and catalase (CAT) and higher levels of antioxidant molecules such as malondialdehyde (MDA) correlate with melanoma disease progression [24,25]. The same correlation was observed with breast cancer cell survival and epithelial mesenchymal transition (EMT) [26]. To coordinate the myriad of functions that ROS have in determining cancer cells’ fate, lncRNAs modulate a fine tuning of the cells’ redox balance.

LncRNAs are noncoding RNA transcripts of 200 or more nucleotides in length of which, until recently, little was known. However, recent studies estimate the number of such molecules to exceed coding mRNAs by as much as 20-fold [27], and the variety of functions that lncRNAs undertake is also significant [28]. Similar to their smaller microRNA (miRNA) counterparts, lncRNAs’ main function is to regulate gene expression, which can be achieved in multiple ways: (1) lncRNAs can induce gene activation and subsequent transcription into mRNA by acting as a liaison between transcription factors and specific gene promoters; (2) lncRNAs can inhibit gene expression by binding to chromatin regions and thus restricting RNA polymerase access to genes within the region; (3) lncRNAs may modulate gene expression by binding and directing enzymes such as histone acetyl transferases or histone methylation enzymes, conversely promoting or inhibiting epigenetic changes; (4) lncRNAs can act as scaffoldings to stabilize multi-subunit complexes that dictate gene expression [29]. Furthermore, in the cytoplasm, lncRNAs can also influence protein activity in a posttranscriptional manner, either by direct interaction with the protein (e.g., allosterically regulating enzymes activities) or by acting as competing endogenous RNAs (ceRNAs) for miRNAs and modifying the translation of mRNA transcripts [30,31].

Thus, not surprisingly, lncRNAs play a pivotal role in cancer cells, acting in different instances either as a part of their arsenal used toward proliferation or as key components in tumor-suppressing mechanisms. Meticulous work has been put into identifying hundreds of lncRNAs over/under expressed in a variety of cancers including but not limited to hepatocarcinoma (HCC) [32,33,34], lung cancer [35,36,37,38], prostate cancer [39,40,41], gastric cancer [42] and renal cell carcinoma (RCC) [43]. 

Recent studies show that various lncRNAs, such as HOTAIR, MALAT1, NRAL and TUG1, play a pivotal role in the regulation of cancer cells’ oxidative stress, highlighting a possible crossroad between lncRNAs and ROS. As such, it is now indicated that multiple key functions of lncRNAs, such as coordinating cancer cells towards proliferation or pro-apoptotic pathways [44], are either triggered or accomplished via variations in the cancer cell’s level of ROS [45]. This is in accordance to the already established property of ROS to drive cancer cells proliferation, while also triggering apoptosis when the level reaches a threshold [46,47,48].

Several lncRNAs could play a role in oxidative stress regulation or could act as responders to oxidative stress. Table 1 summarizes these lncRNAs, along with a brief description of their interaction with the redox system of the cell.

## 2. LncRNAs and the Generation of Mitochondrial ROS in Cancer

The main intracellular source of ROS is the mitochondrial electron transport chain (ETC) [90,91]. A brief description of the ETC and its importance in ROS generation is depicted in Figure 2.

In cancer cells, alterations in the ROS homeostasis can derive either from internal mitochondrial alterations or from external signaling pathways [92,93]. In different cancer cells, the amount of mitochondria-generated ROS is higher than in normal cells [94,95]. One possible explanation is that in phases of hypoxic conditions (such as when the proliferation of tumor cells is high), the availability of O_2_ as an electron acceptor at complex IV is low, causing the accumulation of electrons and increased reduction of complexes I–III. Indeed, several studies have proven that ROS are increased in hypoxic conditions [96,97]. However, given that electrons react quite readily with O_2_ at complex IV (Km < 1 μM), such a low concentration of O_2_ at complex IV should also limit oxygen available for reduction to •O_2_^−^ at other complexes [98].

**Figure 2 ijms-23-10133-f002:**
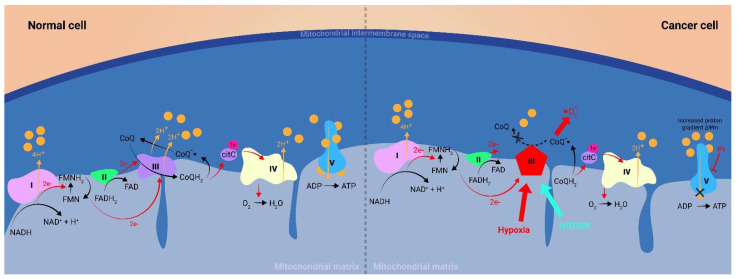
Mitochondrial Electron transport chain (ETC): The ETC consists of five electron exchanging protein complexes bound to the inner mitochondrial membrane. The NADH and FADH2 formed in the tricarboxylic acid chain provide the electrons to complex I and II, respectively, and the electrons are passed on from one complex to another until they reach complex IV where four electrons together with four H+ from the mitochondrial matrix react with oxygen to form two water molecules. The purpose of the ETC is that at each complex–apart from complex, II–H+ are pumped in the intermembrane space (IMS) thus creating a proton gradient between the IMS and the mitochondrial matrix, which polarizes the inner mitochondrial membrane (ΔΨm). This gradient is then used to power the complex V ATP-synthetase mechanism, storing energy in the phosphodiester bound formed between ADP and a molecule of H3PO4. For a more complex depiction of the ETC, see reference [93]. The hypoxic cancer cell environment alongside the increased ΔΨm caused by oxidative phosphorylation inhibitors, such as inhibitory factor 1 (IF1), alter the ETC in such a way that the probability of ROS species formation increases as compared to a normal cell. Certain lncRNAs, such as HOTAIR, stabilize complex III and decrease the probability of ROS formation.

A possible explanation is that hypoxia also affects the conformation of complex III in such a manner that ubisemiquinone (CoQ^−^•) is stabilized at the Qi site of complex III, transforming it into an excellent electron donor and leading to the generation of •O_2_^−^ [93,99]. Furthermore, evidence suggests that lncRNA HOTAIR (HOX transcription RNA) plays a role in the structural stability of complex III. Although HOTAIR has been shown to induce tumorigenesis and metastasis in numerous tumors [100,101], Zheng P. et al. came to the interesting finding that HOTAIR inhibition leads to further depolarization of mitochondrial membrane potential, complex III ultrastructural abnormalities and enhanced intracellular ROS production in human cervical cancer cells [50]. This could indicate that HOTAIR acts as an antioxidant regulator lncRNA and prevents the cancer cell from surpassing the threshold ROS level that would activate apoptosis. This could also explain HOTAIR’s role in chemoresistance observed in lung cancer [51]. It was also observed that lncRNA HOTAIR correlates with the expression of transglutaminase 2 (TGM2) in glioblastoma cells [49]. TGM2 is e key enzyme in maintaining mitochondrial homeostasis under stressful conditions [102], suggesting that its activity plays a part in the stabilization of complex III by HOTAIR. In another recent study, Charlotte Orre et al. proved that lncRNA SAMMSON acts by inhibiting oxidative phosphorylation in MCF-7 doxorubicin-resistant breast cancer cells. This is achieved by inhibiting the transcription and translation of complex I structural proteins, which, in turn, blocks electrons’ passage down the ETC to form ROS [52].

Another theory that could explain the altered mitochondrial ROS metabolism in cancer cells is that the increased mitochondrial membrane potential (ΔΨm) observed in different cancer cell lines is responsible for the increase in ROS production [103,104] thus conferring higher tumorigenicity [105,106,107]. In the context where the ETC is incapacitated by the high ΔΨm, complex I and complex III again seem to be the most important sources of ROS [93,99,108].

## 3. LncRNAs and Nrf2. A Key Point in Cancer Cell’s Oxidative Stress Modulation and Chemoresistance

### 3.1. LncRNAs as Regulators of Nrf2

Nuclear factor (erythroid-derived 2)-like 2 (Nrf2)—a nuclear transcription factor for cellular anti-oxidation responses—positively correlates with various cancers’ progression and chemoresistance [109,110]. Increasing evidence suggests that lncRNAs play a key part in regulating Nrf2 expression [111]. This can be achieved either directly, by epigenetic alterations and posttranslational interactions with Nrf2, or indirectly, by influencing the expression of other molecules that are important regulators of Nrf2 (Figure 3).

LncRNA TUG1 post-translationally potentiates the Nrf2 effect in esophageal squamous cell carcinoma (SCC) cells, by directly binding to Nrf2 protein and elevating its expression [54]. 

LncRNA SLC7A11-AS1 confers chemoresistance in pancreatic adenocarcinoma cells by augmenting the expression of Nrf-2. By interacting with the F-box motif of β-TRCP1, the exon 3 of SLC7A11-AS1 prevents the recruitment of β-TRCP1 to the SCFβ-TRCP E3 complex, which would otherwise initiate ubiquitination and proteasomal degradation of Nrf2 in the nucleus [56,112]. 

Another study carried out by Luo P. et al. found that small interfering RNA (siRNA) targeting lncRNA MIR4435-2HG significantly reduced cisplatin resistance in colon cancer, which was attributed to the decrease in Nrf2 mRNA observed using qPCR [57].

Cisplatin-resistant HCC cells also presented specific lncRNA signatures in microarray and qPCR analyses. Following Luciferase reporter assays, it was confirmed that Nrf2 mRNA was targeted and inactivated by a miRNA-miR340–5p- and that Nrf2-associated lncRNA (NRAL) could function as a ceRNA for Nrf2 by binding to miR340–5p, thereby freeing Nrf2 antioxidant effects [58]. 

Another way by which lncRNAs influence the expression of Nrf2 is by modulating Kelch-like ECH-associated protein 1 (KEAP1) activity, a cytosolic protein that prevents Nrf2 translocation to the nucleus and accomplishing its role as a transcription factor. One example is lncRNA KRAL, which—similar to NARL lncRNA—acts as a ceRNA by competitively binding miR-141 and thereby freeing KEAP1. Given that KRAL is downregulated in 5-fluorouracil-resistant HCC cells, this leads to increased levels of Nrf2, thus also helping the cell mitigate its increased ROS level [59]. On the other hand, lncRNA MALAT-1 was observed to be overexpressed in different cancer cell lines [113] and to lower KEAP1 activity [114]. 

Thus, the decrease in KRAL and the increase in MALAT1 regulate KEAP1 and along with the overexpression of TUG1, NRAL and MIR4435-2HG observed in cancer cells, converge to increase the antioxidant potential of the cell, permitting it to tolerate higher oxidative stress, and once again substantiating the important influence of lncRNA on ROS in cancer.

### 3.2. LncRNAs Regulated by Nrf2

Inversely, the idea that Nrf2 antioxidant functions could be accomplished by lncRNAs has recently appeared.

One of the first lncRNAs that was identified to be regulated by Nrf2 was lncRNA SCAL1 (smoke and cancer-associated lncRNA 1). Thati P. et al. demonstrated that siRNA knockdown of Nrf2 caused a significant decrease in SCAL1, with subsequent increased susceptibility to oxidative stress caused by cigarette smoke [67].

LncRNA TUG1, which we previously noted increases the expression of Nrf2, has also been shown to be positively regulated by Nrf2 in urothelial carcinoma of the bladder cells [54], thus completing a possible positive feedback loop that sustains the antioxidant response of cancer cells. 

Another study conducted by Gao M. et al. [69], validated the binding of Nrf2 transcription factor to lncRNA ODRUL (Osteosarcoma doxorubicin-resistance related upregulated lncRNA) gene promoter as a response to oxidative stress. In this case, ODRUL-mediated the pro-apoptotic effect of NFf2 by diminishing anti-apoptotic Bcl-2 protein levels. 

Furthermore, Moreno Leon L. et al. [73] found that pharmacological inhibition of NF-κB-dependent transcription or siRNA-mediated depletion of Nrf2 downregulates the hypoxia dependent expression of lncRNA NLUCAT1. To explore the impact of this NLUCAT 1 downregulation without the bias of other lost functions due to inhibition of NF-κB or NRF2 transcription, they used CRISPR-Cas9 gene editing to invalidate NLUCAT-1 gene and found that NLUCAT-1 knockdown lung adenocarcinoma cells showed lower proliferation rates, higher level of ROS and thus an increased susceptibility to cisplatin-induced apoptosis. It follows that NLUCAT-1 represents another lncRNA induced by NRF2 with an important role in the antioxidant response of cancer cells.

The importance of such antioxidant responses conducted by lncRNAs is uncanny as it confers the property of chemoresistance to drugs or radiation therapy that often act by increasing the cell oxidative stress, triggering apoptosis. 

## 4. A LncRNA Perspective on the Warburg Effect. Tipping the Redox Balance

Now generally accepted as a core hallmark of cancer, metabolic reprogramming (Warburg effect) shifts the use of glucose from an energy-producing molecule, to a biomass source, used in the synthesis of essential proteins and nucleic acids for the rapidly dividing cancer cells [115]. This is attained by redirecting a large part of the glucose intake of the cell away from oxidative phosphorylation and towards the glycolytic pathway. While in normal cells that are not actively growing, this process happens only under hypoxic conditions, in the cancer cell, through activation of oncogenes such as MYC and PI3K/AKT signaling, this happens even in normoxic conditions, hence the term aerobic glycolysis [115]. Recently, certain lncRNAs have emerged as important regulators of the metabolic reprogramming seen in cancer cells (Figure 4). LncRNA SOX-2-OT promotes HCC invasion and metastasis by orchestrating such a metabolic shift. Lyang Y. et al. [70] found that the overexpressed SOX-2-OT acts by inducing the PKM2 isoform of the enzyme PK (Pyruvate Kinase), which catalyzes the transformation of phosphoenolpyruvate into pyruvic acid, a rate limiting step of glycolysis. Despite its association with highly proliferative and metabolic active cells, the PKM2 isoform has lower intrinsic enzymatic activity than its PKM1 counterpart. Moreover, PKM2 is also sensible to particular inhibitory factors such as ROS formed in the ETC [116,117,118]. As such, the glucose metabolism will be partly shifted towards the pentose phosphate pathway form which NADPH will be formed as a byproduct. Consequently, NADPH can be used as an electron donor for most of the antioxidant mechanisms the cancer cell is reliant on to keep its high level of ROS in check [119,120].

Moreover, LncRNA that is highly upregulated in liver cancer (HULC) has been shown to sustain metabolic reprogramming and tumorigenesis in HCC by inducing the PKM2 isoform [71,121]. Multiple mechanisms have been proposed. One study [71] used a combination of tobramycin affinity purification and mass spectrometric analysis (TOBAP-MS) and found that HULC directly binds PKM2, augmenting its interaction with fibroblast growth factor receptor 1 (FGFR1). This causes increased enzyme phosphorylation with subsequent reduced formation of the tetrameric, more active form [122]. Furthermore, HULC can promote the expression of the PKM2 isoform via inhibition of the tumor suppressor phosphatase and tensin homolog (PTEN). Mechanistically, Xin X et al. [72] found that HULC inhibited the expression of PTEN on the translational level, but not on the transcriptional level, leading to the supposition that HULC acts by promoting cancer cell autophagy. This supposition was later confirmed by the observation that HULC could not alter the expression of PTEN after administration of MG132 (ubiquitin–proteasome inhibitor). Taken together, HULC promotes hepatocarcinogenesis by augmenting autophagy, depressing the PTEN tumor suppressor and inducing the PKM2 isoform, thus altering the redox potential of the cancer cell.

## 5. ROS and LncRNAs Determine Cancer Cell Fate. Possible Therapeutic Strategies

Besides being important regulators of the antioxidant response in cancer cells, lncRNAs take part in different ROS-dependent pathways essential for sustaining the hallmarks of cancer.

### 5.1. ROS, lncRNAs and Cell Proliferation

MALAT1 (metastasis-associated lung adenocarcinoma transcript 1, also known as NEAT2 or nuclear enriched abundant transcript 2) is directly correlated with the aggressiveness and mortality of lung adenocarcinoma [29,113,114]. The correlation between ROS and MALAT1 may be attributed to the effect of ROS on decreasing the degradation of hypoxia-inducible factor (HIF 1) α, a key transcriptional factor of MALAT-1 [123]. The preservation of HIF-1-α is achieved by the ROS-mediated inactivation of prolyl hydroxylase (PHDs) [124,125], which would otherwise determine the hydroxylation of specific prolyl residues in the HIF-1-α molecule, resulting in binding to the von Hippel–Lindau (VHL) protein of the E3 ubiquitin ligase complex and subsequent degradation via the polyubiquitination/proteasomal degradation pathway [126]. However, the role of MALAT1 in cell proliferation has been recently contradicted. While several studies have suggested that MALAT1 has a pro-growth effect on tumor cells [60,61,127] and that MALAT1 depleted cells undergo G1/S cell cycle arrest [62], recent studies postulate a tumor suppressive role of MALAT1 [63,64,65]. Nevertheless, MALAT1 has an important role in tumor progression and ROS can modulate its expression through HIF-1-α.

HIF-1-α also promotes the transcription of other lncRNAs correlated with tumorigenesis, such as HOTAIR [128], NUTF2P3-001 [129], UCA1 [130] and BX111 [131], but their direct correlation with the level of ROS has not yet been documented. 

LncRNA XIST is another oncogenic lncRNA correlated with non-small cell lung cancer (NSCLC) proliferation, invasion, and metastasis [74,75]. The level of expression of XIST in NSCLC was also found to influence the cancer cells’ oxidative stress [76]. As such, Liu J et al. concluded that XIST knockdown NSCLC cells undergo pyroptosis due to a significant increase in the intracellular level of ROS. Treatment of such cells with N acetylcysteine (NAC), a potent antioxidant due to its thiol group, prevented pyroptosis. In researching why the loss of XIST distorts cancer cells’ oxidative stress control, they have found that it can act as a sponge for miR-335, inducing an increase in SOD2 expression, a crucial enzyme in the antioxidant process of cells, transforming •O_2_^−^ to H_2_O_2_. Another study on osteosarcoma cancer cells found that XIST can also sponge miR-153, leading to oxidative stress-induced proliferation and invasion by potentiating Snail expression, one of the master regulators of EMT [77].

### 5.2. ROS, lncRNAs and Cell Death

Growth arrest-specific transcript 5 lncRNA (GAS-5) has been known to act as a tumor suppressor and to be notably downregulated in various cancers such as breast [83], prostate [84] and gastric cancer [85]. However, GAS-5 effect on the cellular level of ROS has been a subject of controversy. Chen et al. [86] used dihydroethidium (DHE) fluorescent staining and flow cytometry for the measurement of •O_2_^−^ and qRT-PCR for the measurement of GAS5 in A375 and SK-Mel-28 human MM cell lines. Their results suggested that GAS5 expression was inversely proportional to the intracellular ROS level, resulting in disease progression and increased cell viability in GAS5 deficient cells, which is in accordance with previous studies on the effect of ROS on MM cell lines [132,133,134,135]. However, contrary results were obtained by Xu U. et al. [87], finding that A375 cells with low expression of GAS5 exhibited suppressed oxidative stress. In addition, studies assessing the effect of GAS5 in non-cancerous cells have also found that it can decrease the level of intracellular ROS by sponging different miRNAs (miR 452-5p, miR135a) [88,89]. It transpires that the GAS5 effect of inhibiting tumor growth can be partly attributed to its effect on the level of cellular ROS, but further research is needed to conclude whether this is achieved by augmenting ROS expression and promoting apoptosis or by mitigating the pro-growth effect of ROS.

Moreover, the apoptosis mediated by ROS p53 activation seems to be dependent on the transcription of lncRNAs. LincRNA-p21 (long intergenic noncoding RNa-p21) is a long interfering noncoding RNA that is under-expressed in different cancers [78]. Huarte M. et al. [136] indicated that RNAi-mediated knockdown of lincRNA-p21 severely decreases apoptosis. To further reinforce the importance of the ROS/p53/lincRNa-p21 pathway in cancer cell apoptosis, an impact of ROS inhibition on lincRNa-p21 should be assessed. LncRNA NEAT1 is another important cell death regulator that lays downstream of p53 in the ROS-induced ferroptosis of HCC. By sponging miR-362-3p, NEAT1 promoted Myo-inositol oxygenase (MIOX) expression, resulting in increased ROS production and ferroptosis [79]. Inversely, lncRNA ROR can inhibit apoptosis in liver cancer cells by suppressing the p-53 response to ROS exposure [80]. LncRNA NORAD increases the expression of autophagy-related genes ATG-5 and ATG-12, which reduces oxidative stress and DNA damage in gastric cancer cells [81,137]. In breast cancer cells, the lncRNA linc00963 sponges miR324-3p, an inhibitor of ACK1, a non-receptor-tyrosine-kinase that drives tumor progression [82]. In nasopharyngeal carcinoma, miR324-3p has also been shown to inhibit WNT2B, part of the WNT family of protein ligands, which are important in cancer cells’ stemness [138,139]. As such, linc00963 allows the cell to mitigate the DNA damage induced by ROS and avoid apoptosis.

### 5.3. The Road Forward. Possible Therapeutic Strategies

Given how quintessential lncRNAs are proving to be for oxidative stress regulation and how important ROS are to sustain cancer cells’ hallmarks, it follows that therapies targeting these lncRNAs are increasingly studied.

ROS has long been a target of the arsenal used by medical practitioners against cancer. Drugs such as cisplatin, anthracyclines and cyclophosphamide [140] all attain their chemotherapeutic effect, at least partly, by dysregulating the redox balance in cancer cells and leading the cell to oxidative stress-triggered apoptosis. For example, cisplatin is known to increase the intracellular level of ROS through disruption of mitochondrial membrane potential [141,142,143], while anthracyclines appear to cause direct •O_2_^−^ formation trough the metabolism of the molecule due to its quinone and hydroquinone groups [144,145]. Moreover, it is indicated that both drugs could also cause depletion of cells’ glutathione reserves, thus preventing cancer cells’ antioxidant machinery from mitigating the increase in ROS [146,147]. Even external therapies such as radiotherapy also exert a cytotoxic effect through the ROS generated by the radiolysis of extracellular water [148,149]. However, one of the major disadvantages of these therapies is the toxicity exerted by the same mechanisms on healthy cells. For example, both the neurotoxicity [150,151,152,153] and nephrotoxicity [154,155] induced by cisplatin have been demonstrated to be correlated with the level of generated ROS.

In the search for novel therapeutic targets, lncRNAs have emerged as valid contenders due to their cancer cell line specificity and low level of expression in other cells. A perfect lncRNA target from a therapeutic standpoint would check 4 essential boxes: predominant expression of a single isoform (insignificant alternative splicing), cancer cell restricted expression, specific function, and having a highly conserved structure throughout species to allow for extrapolation from in vivo studies [156,157]. Although no single lncRNA satisfies all the above conditions, several intriguing targets have been identified. Multiple methods for such targeting have been proposed. Firstly, silencing lncRNA gene transcription using systems, such as clustered regularly interspaced short palindromic repeats (CRISPR)/Cas9 or zinc finger nucleases (ZFNs), has been successfully utilized in vitro, resulting in suppression of the targeted lncRNA effect [158,159]. Nonetheless, given that lncRNAs can share promoters with other coding or non-coding genes, using the CRISPR/Cas9 system for targeting lncRNAs comes with the risk of inadvertently deregulating neighboring genes [160]. Secondly, disrupting the interaction between a lncRNA and its specific target would, in theory, alter the transcription of the genes normally regulated by the interaction. For example, using a newly designed molecule named AC1N0D4Q, Ren et al. managed to block the interaction between lncRNA HOTAIR and the EZH2 catalytic subunit of the PCR2 complex, which resulted in in a reduction of breast cancer cell migration and invasion in both in vitro and animal models [161]. However, controversy still exists over the specificity of some of these lncRNA-gene regulating complex interactions, given that they function based on scaffolding rather than complementarity, as RNA-DNA interactions do [162,163,164,165]. Thirdly, reducing the level of expression of lncRNAs by degradation of the non-coding transcripts per se is another approach that has proved efficient in several studies. This can be achieved by siRNA or antisense oligonucleotides (ASO). Indeed, administration of MALAT1 targeting ASOs reduced metastasis in both human lung cancer xenograft models [60] and mouse mammary tumor models [166], while siRNA knockdown of MALAT1 reduced cell motility in vitro models [167]. Moreover, clinical trials using these methods have also rendered positive results. The nanocomplex delivery of MALAT1 targeting siRNA in glioblastoma patients increased the sensitivity of the tumor to temozolomide (TMZ) treatment [168], while a novel ASO drug against non-coding mitochondrial RNA demonstrated good tolerance in a phase one clinical study [169,170].

## 6. Conclusions and Further Perspectives

The dysregulation of oxidative stress is an essential aspect that allows cancer cells to take an advantage over neighboring cells. However, ROS is a double-edged sword, as excess oxidative stress induces growth inhibition and cell death. Conversely, cancer cells have developed elaborate mechanisms to calibrate the level of oxidative stress. LncRNAs are indispensable in this process, accommodating the increased level of ROS through multiple mechanisms. Several lncRNAs regulate the ROS homeostasis at the mitochondrial level, influencing the level of ROS generated in the ETC. Furthermore, by inducing Nrf2 gene expression, acting as ceRNA for Nrf2 mRNA targeting miRNA, preventing Nrf2 protein cytoplasmatic and nuclear inactivation and acting as effectors in the Nrf2-induced antioxidant response, lncRNAs sustain the activity of the main transcription factor of antioxidant response. LncRNAs also sustain the Warburg effect in cancer cells by promoting PKM2 isoform expression, another effective mechanism that perpetuates the dysregulated oxidative balance. Moreover, lncRNAs can act in synergy with ROS in orientating the cancer cell toward proliferation or apoptosis by modulating the expression of numerous important genes in the ROS pathways (e.g., TP53, HIF-1-α).

The specificity of an unstable redox balance and patterns of lncRNA expression to cancer cells offer a great therapeutic opportunity in the fight against cancer. However, although some clinical trials targeting lncRNAs have shown encouraging results, the majority of the research is still in the preclinical phase and further investigation and development through in vitro studies and animal models is required before lncRNA-targeted medicine can become a mainstream approach.

## Figures and Tables

**Figure 1 ijms-23-10133-f001:**
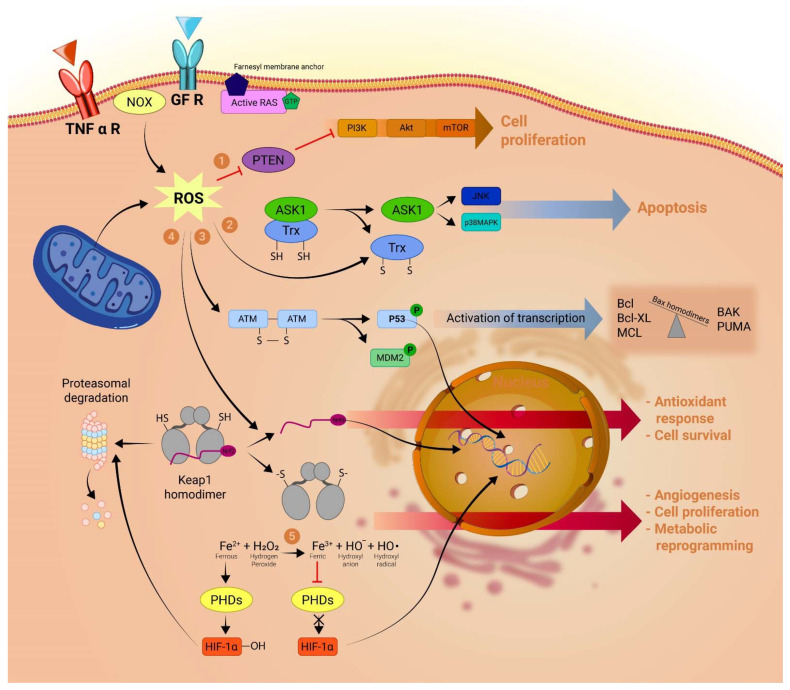
ROS functions in cancer cells: The production of ROS is elevated in tumor cells as a consequence of increased metabolic rate, gene mutation, extracellular growth factor signaling and relative hypoxia. This increase in the baseline ROS level plays a crucial role in determining cancer cells’ fate: (1) ROS can drive cancer cell proliferation by inactivating PTEN, a tumor suppressor that acts as a brake on the pro-growth PI3K/Akt pathway. (2) Inversely, increased ROS can have a nefarious effect on the cancer cell by activating apoptotic pathways. As such, oxidative stress may lead to the formation of disulfide bonds in the thioredoxin molecule, which results in the activation of the kinase Ask1 and subsequent activation of key apoptotic transcription factors like JNK and p38. (3) A similar process occurs in the case of p53 activation, where an ROS-triggered formation of disulfide bonds at the Cys 2999 residues of ATM homodimer results in activation and subsequent phosphorylation of p53 and its inhibitory molecule MDM2. P53 alongside JNK, p38, and other transcription factors will shift the balance towards apoptosis by expressing proteins, such as Bax homodimers, BAK and PUMA, to the detriment of antiapoptotic proteins such as Bcl-2, Bcl-XL or MCL. (4) ROS can prevent the proteasomal degradation of transcription factor Nrf2 by dissociating the KEAP1-Nrf2 complex and thus allowing Nrf2 to translocate to the nucleus and activate antioxidant pathways that can mitigate the proapoptotic effect of ROS. (5) By oxidizing iron from its ferrous (Fe^2+^) to its ferric (Fe^3+^) state, ROS dampens the enzymatic activity of PHDs which would normally hydroxylase the HIF-1α protein causing its proteasomal degradation. The non-hydroxylated form of HIF-1α has a major role in promoting cancer cell survival, angiogenesis and metabolic reprogramming under hypoxic conditions. PTEN, phosphatase and tensin homolog; Ask1, apoptosis signal-regulating kinase 1; ATM, ataxia-telangiectasia mutated; MDM2, mouse double minute 2 homolog; Bax, Bcl-2-associated protein X; Bcl-2, B-cell lymphoma 2; KEAP1, Kelch-like ECH-associated protein 1; PHDs, prolyl hydroxylase domain enzymes; HIF-1α, hypoxia-inducible factor α.

**Figure 3 ijms-23-10133-f003:**
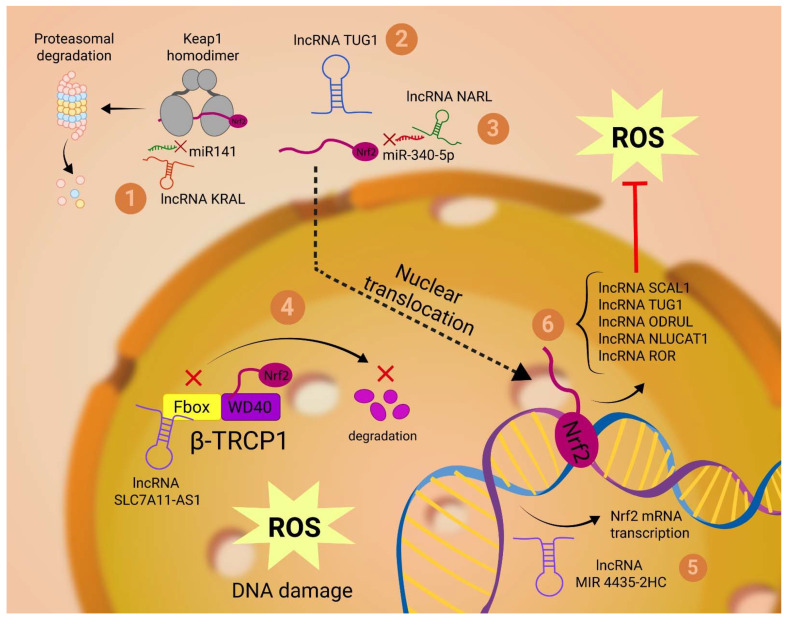
Nrf2-associated lncRNAs: As regulators of Nrf2, lncRNAs can act both at the cytoplasmatic and the nuclear level. In the cytoplasm, lncRNA KRAL and lncRNA NARL act as ceRNAs for miR 141 and miR-340-5p, respectively. While KRAL promotes Nrf2 proteasomal degradation by freeing up the KEAP 1 homodimer (1), NARL prevents Nrf2 inactivation in the cytoplasm and facilitates its translocation to the nucleus (2). At the cytoplasmatic level, lncRNA TUG1 directly binds to the Nrf2 protein and potentiates its activity (3). In the nucleus, lncRNA SLC7A11-AS1 prevents the recruitment of β-TRCP1 to the SCFβ-TRCP E3 complex, which would otherwise initiate ubiquitination and proteasomal degradation of Nrf2 in the nucleus (4). LncRNA MIR4435-2HG induces Nrf2 gene expression and subsequent transcription into mRNA (5). Nrf2 induces the transcription of multiple lncRNAs that regulate the level of ROS (6).

**Figure 4 ijms-23-10133-f004:**
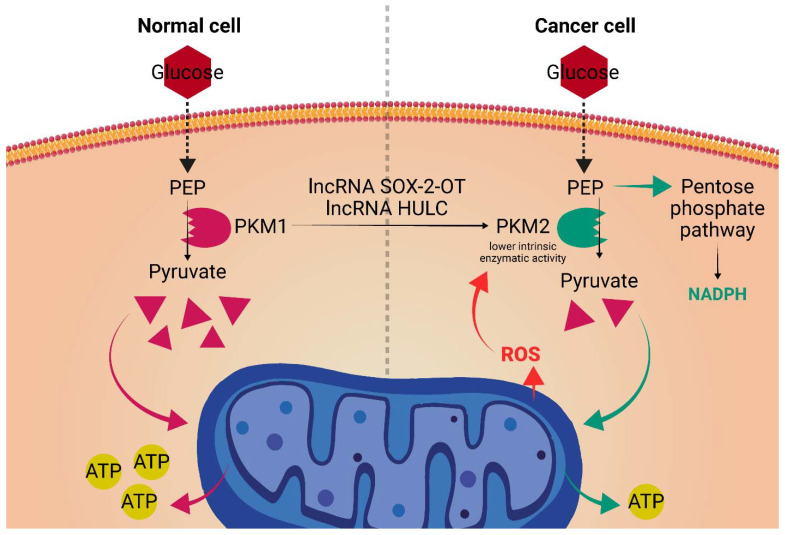
LncRNAs’ role in the Warburg effect: SOX-2-OT and HULC lncRNAs carry out an antioxidant effect by inducing the PKM2 isoform of pyruvate kinase. Given its lower intrinsic enzymatic activity and its susceptibility to inhibition by ROS, the expression of the PKM2 isoform will allow for the synthesis of key antioxidant molecules such as NADPH. PEP, phosphoenolpyruvate; PKM1, pyruvate kinase M1 isoform; PKM2, pyruvate kinase M2 isoform. LncRNAs’ role in the Warburg effect: SOX-2-OT and HULC lncRNAs carry out an antioxidant effect by inducing the PKM2 isoform of pyruvate kinase. Given its lower intrinsic enzymatic activity and its susceptibility to inhibition by ROS, the expression of the PKM2 isoform will allow for the synthesis of key antioxidant molecules such as NADPH. PEP, phosphoenolpyruvate; PKM1, pyruvate kinase M1 isoform; PKM2, pyruvate kinase M2 isoform.

**Table 1 ijms-23-10133-t001:** Important lncRNAs in the regulation of oxidative stress and their impact on cancer cells.

LncRNA	Expression in Cancer Cells	Role in Oxidative Stress Regulation	ROS Level	Impact on Cancer Cells	Refs.
lncRNA HOTAIR	↑ (increased) in lung cancer	confers structural stability to complex III	↓(decreased)	↑ chemoresistance and survival	[49,50,51]
	↑ in glioblastoma ↑ in cervical cancer			↑ invasion and metastasis	
lncRNA SAMMSON	↑ in breast cancer	inhibits complex I protein transcription and translation	↓	↑ chemoresistance and survival	[52]
lncRNA TUG1	↑ in oesophageal SCC	↑ antioxidant response by potentiating Nrf2 expression expression is also stimulated by Nrf2	↓	↑ chemoresistance and survival	[53,54,55]
	↑ in urothelial carcinoma of the bladder			↑ cell proliferation ↓ apoptosis	
lncRNA SLC7A11-AS1	↑ in pancreatic adenocarcinoma	prevents proteosomal degradation of Nrf2	↓	↑ chemoresistance	[56]
lncRNA MIR4435-2HG	↑ in colon cancer	increases Nrf2 expression	↓	↑ chemoresistance ↑ cell proliferation	[57]
lncRNA NRAL	↑ in HCC	acts as an ceRNA by binding miRNA-miR340–5p and thereby freeing Nrf2 antioxidant effects	↓	↑ chemoresistance	[58]
lncRNA KRAL	↓ in HCC	acts as an ceRNA by binding miRNA miR-141 thereby freeing KEAP1 and preventing nuclear translocation of Nrf2	↑	↑ chemoresistance	[59]
lncRNA MALAT1	↑ in NSCLC	HIF 1 α dependent transcription	↓	↑ cell survival and proliferation	[60,61,62,63,64,65]
		↓ KEAP1 activity		possible tumor suppressive role	
lncRNA SCAL1	↑ after exposure to cigarette smoke	effector of Nrf2 antioxidant response	↓	↑cryoprotection against cigarette smoke–induced toxicity	[66,67]
	↑ in NSCLC			↓ apoptosis in NSCLC	
lncRNA ODRUL	↑ in osteosarcoma	mediates Nrf2 pro-apoptotic effects	↓	↑ doxorubicin resistance	[68,69]
lncRNA SOX-2-OT	↑ in HCC	inducing the PKM2 isoform of the enzyme PK	↓	metabolic reprogramming	[70]
				↑ invasion and metastasis	
lncRNA HULC	↑ in HCC	inducing the PKM2 isoform of the enzyme PK	↓	metabolic reprogramming	[71,72]
lncRNA H19	↑ in HCC	decreases SOD activity via MAPK/ERK pathway	↓	↑ cell viability	[33]
				↓ cell apoptosis	
				↑ chemoresistance	
lncRNA NLUCAT	↑ in lung adenocarcinoma	effector of Nrf2 antioxidant response	↓	↑ cell proliferation	[73]
				↓ cisplatin susceptibility	
lncRNA XIST	↑ in NSCLC ↑ in osteosarcoma	acts as a sponge for miR-335, inducing an increase in SOD2 expression acts as a sponge for miR-153, increasing Snail expression	↓	prevents pyroptosis ↑ cell proliferation ↑ EMT	[74,75,76,77]
lincRNA-p21	↓ in NSCLC	induces apoptosis as a result of ROS-mediated p53 activation	↑	↓chemoresistance ↓cell survival	[78]
	↓ in HCC				
	↓ in breast cancer				
lncRNA NEAT1	↑ in HCC	induces apoptosis as a result of ROS-mediated p53 activation		↑ apoptosis	[79]
lncRNA ROR	↑ in HCC	Inhibits p53 activity		↓ cell apoptosis	[80]
lncRNA NORAD	↑ in gastric cancer	increases the expression of autophagy related genes ATG-5 and ATG-12	↓	↑cell survival	[81]
lincRNA 00963	↑ in breast cancer	sponges miR324-3p	↓	↑cell survival ↑ chemoresistance	[82]
lncRNA GAS-5	↓ in breast cancer	inhibits NOX4 protein expression Inhibits G6PD protein expression	↑/↓	↓ tumor suppressor effect	[83,84,85,86,87,88,89]
	↓ in prostate cancer				
	↓ in gastric cancer			↑cell survival and proliferation	
	↓ in melanoma				

## Data Availability

Not applicable.

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
