# Peer review of "Reactive Oxygen Species and Long Non-Coding RNAs, an Unexpected Crossroad in Cancer Cells"

_ijms, 2022, doi:10.3390/ijms231710133_

Round 1

Reviewer 1 Report

The manuscript "Reactive oxygen species and long noncoding RNAs. An unexpected crossroad in cancer cells" is intriguing, explained from the basics, and the figures are informative and elaborative. The article explored how long non-coding RNAs (lncRNA) regulate oxidative stress in cancer cells. However, I have the following suggestions for improvement of the manuscript:

1.     May change the title to "Reactive oxygen species and long non-coding RNAs, an unexpected crossroad in cancer cells"

2.     The figures look hazy. The orange color is not visible in Fig. 1; try to improve the quality. It should also be color coordinated for more informative and better aesthetics.

3.     There are many grammatical and spelling mistakes e.g. tumorigenesis line 111; correlate line 118,155,315,318; line 142; figure line 159, 231; determine line 269; an electron line 243; change that to than line 239; reprogramming at many places 230, 232; fate line 282; proteasomal degradation line 292, 296; rephrase 253-254 etc

Hyphen is inconsistent and missing in many compound modifiers, e.g., cancer-associated, RNAi-mediated.

Check the whole manuscript to omit these mistakes and others.

4.     The manuscript needs a minor change in the style of English; it sounds more like a book chapter instead a review article.

5.     Cite and discuss recently published articles

Zuo, J., Zhang, Z., Li, M. et al. The crosstalk between reactive oxygen species and noncoding RNAs: from cancer code to drug role. Mol Cancer 21, 30 (2022). https://doi.org/10.1186/s12943-021-01488-3

 Wu YZ, Su YH, Kuo CY. Stressing the Regulatory Role of Long Non-Coding RNA in the Cellular Stress Response during Cancer Progression and Therapy. Biomedicines. 2022;10(5):1212. Published 2022 May 23. doi:10.3390/biomedicines10051212

 Orre C, Dieu X, Guillon J, Gueguen N, Ahmadpour ST, Dumas JF, Khiati S, Reynier P, Lenaers G, Coqueret O, Chevrollier A, Mirebeau-Prunier D, Desquiret-Dumas V. The Long Non-Coding RNA SAMMSON Is a Regulator of Chemosensitivity and Metabolic Orientation in MCF-7 Doxorubicin-Resistant Breast Cancer Cells. Biology (Basel). 2021 Nov 9;10(11):1156. doi: 10.3390/biology10111156. PMID: 34827149; PMCID: PMC8615054.

Author Response

Dear esteemed referee,

Thank you for allowing us to submit a revised draft of the manuscript “Reactive oxygen species and long non-coding RNAs, an unexpected crossroad in cancer cells” to IJMS special issue “Role of ncRNAs Classes as Biomarkers for Diagnostic and Prognosis in Cancer 2021 . We appreciate the time and effort that you have dedicated to providing your valuable feedback on our manuscript. We have been able to incorporate changes to reflect most of the insightful suggestions provided.  The changes are inserted with track changes in the revised manuscript and are also presented below. When refereeing to a line in the revised manuscript, the view with “show insertion and deletions” is used.

  1. May change the title to "Reactive oxygen species and long non-coding RNAs, an unexpected crossroad in cancer cells"

Thank you for your suggestion, we changed the title accordingly

“Reactive oxygen species and long non-coding RNAs, an unexpected crossroad in cancer cells”

  1. The figures look hazy. The orange color is not visible in Fig. 1; try to improve the quality. It should also be color coordinated for more informative and better aesthetics.

Response: We agree with the reviewer’s assessment. Accordingly, all the figures have been updated aesthetically. The information in the figures, however, hasn’t changed.

  1.      There are many grammatical and spelling mistakes. Hyphen is inconsistent and missing in many compound modifiers, e.g., cancer-associated, RNAi-mediated.

Check the whole manuscript to omit these mistakes and others.

Response: Thank you for pointing this out. The aforementioned mistakes were corrected, and the manuscript was rechecked for any other spelling or grammatical mistakes.

-tumorgenesis (line 111) was changed to tumorigenesis (line 202)

-corelate (line 118,155,315,318) was changed to correlate (209,238,392,395,462)

-fiugre (line 159,231) was changed to figure (line 242,323)

-a electron (line 243) was changed to an electron (line 335)

-that (line 239) was changed to than (line 332)

-reprograming was changed to reprogramming throughout the manuscript

-fait (line 282) was changed to fate (line 79)

-proteosomal (line 292,296) was changed to proteasomal (line 90, 94),fig 1,3

-lines 253-254 were rephrased from “There are also other lncRNAs that can influence the expression of PKM2 isoform and favor tumorigenesis via metabolic reprograming. LncRNA highly upregulated in liver cancer (HULC) does so through multiple possible pathways” to “Moreover, LncRNA highly upregulated in liver cancer (HULC) has been shown to sustain metabolic reprogramming and tumorigenesis in HCC by inducing the PKM2 isoform” (line 346-348)

  1. The manuscript needs a minor change in the style of English; it sounds more like a book chapter instead a review article.

Response: We agree with the reviewer’s assessment. Accordingly, several statements were rephrased, e.g.

- “as a useful defense mechanism against the host’s foe” (line 87-88) was changed to “key components in tumor-suppressing mechanisms” (line 126-127)

- “As it transpires from our previous discussion, certain lncRNAs are important regulators of the antioxidant response in cancer cells, thus preventing apoptosis and maintaining cell survival and chemoresistance” (line 270-272) was changed to “Besides being important regulators of the antioxidant response in cancer cells, lncRNAs take part in different ROS dependent pathways essential for sustaining the hallmarks of cancer” (line 365-367)

- “by virtue of its lower intrinsic enzymatic activity” (line 248) was changed to “Given its lower intrinsic enzymatic activity” (line 342)

  1. Cite and discuss recently published articles

Thank you for sharing these recently published articles. Although many of the lncRNAs discussed in the reviews were already presented in our paper, we found interesting insights that we managed to include in our paper:

Zuo, J., Zhang, Z., Li, M. et al. The crosstalk between reactive oxygen species and noncoding RNAs: from cancer code to drug role. Mol Cancer 21, 30 (2022). https://doi.org/10.1186/s12943-021-01488-3

This was an excellent review on the possible therapeutic strategies involving ncRNAs important in oxidative stress regulation. After carefully reading the paper and its’ discussed articles of interest for the scope of our paper, we made the following modifications

-  we have included linc00963 as a new lncRNA in our discussion of the lncRNAs that regulate apoptosis in cancer cells: “In breast cancer cells, the lncRNA linc00963 sponges miR324-3p, an inhibitor of ACK1, a non-receptor-tyrosine-kinase that drives tumor progression[81]. In nasopharyngeal carcinoma, miR324-3p has also been shown to inhibit WNT2B, part of the WNT family of protein ligands, which are important in cancer cells’ stemness[138,139]. As such, linc00963 allows the cell to mitigate the DNA damage induced by ROS and avoid apoptosis.”(line 439-443)

-we have expanded our discussion of the studies on therapeutic strategies using antisense oligonucleotides or siRNA to target lncRNAs: “[164], while siRNA knockdown of MALAT1 reduced cell motility in in vitro models[165]. Moreover, clinical trials using these methods have also rendered positive results. Nanocomplex delivery of MALAT1 targeting siRNA in glioblastoma patients increased the sensitivity of the tumor to temozolomide (TMZ) treatment[166], while a novel ASO drug against non-coding mitochondrial RNA demonstrated good tolerance in a phase one clinical study[167,168]. “(line 489-495)

 Wu YZ, Su YH, Kuo CY. Stressing the Regulatory Role of Long Non-Coding RNA in the Cellular Stress Response during Cancer Progression and Therapy. Biomedicines. 2022;10(5):1212. Published 2022 May 23. doi:10.3390/biomedicines10051212

This review stressed the importance of numerous lncRNAs in the cancer cell’s response to stress. From the ones that were not discussed in our paper, we included three new lncRNAs:

-lncRNA NEAT1: “LncRNA NEAT1 is another important cell death regulator that lays downstream of p53 in the ROS induced ferroptosis of HCC. By sponging miR-362-3p, NEAT1 promoted Myo-inositol oxygenase (MIOX) expression, resulting in increased ROS production and ferroptosis[78] “ (line 432-434)

-lncRNA ROR: “Inversely, lncRNA ROR can inhibit apoptosis in liver cancer cells by suppressing the p-53 response to ROS exposure[79].” (line 434-435)

-lncRNA NORAD: “LncRNA NORAD increases the expression of autophagy related genes ATG-5 and ATG-12, which reduces oxidative stress and DNA damage in gastric cancer cells[80,137]“ (line 437-438)

We have also added another study to our discussion of the lncRNA XIST: “Another study on osteosarcoma cancer cells found that XIST can also sponge miR-153, leading to oxidative stress-induced proliferation and invasion by potentiating Snail expression, one of the master regulators of EMT[76]. ” (line 403-406)

 Orre C, Dieu X, Guillon J, Gueguen N, Ahmadpour ST, Dumas JF, Khiati S, Reynier P, Lenaers G, Coqueret O, Chevrollier A, Mirebeau-Prunier D, Desquiret-Dumas V. The Long Non-Coding RNA SAMMSON Is a Regulator of Chemosensitivity and Metabolic Orientation in MCF-7 Doxorubicin-Resistant Breast Cancer Cells. Biology (Basel). 2021 Nov 9;10(11):1156. doi: 10.3390/biology10111156. PMID: 34827149; PMCID: PMC8615054.

Orre C. and colleagues have conducted an impressive study on the role and mechanisms of lncRNA SAMMSON in breast cancer. We have decided to include this paper in our discussion on the roles of lncRNAs in mitochondrial ROS generation: “In another recent study, Charlotte Orre et al. proved that lncRNA SAMMSON acts by inhibiting the oxidative phosphorylation in MCF-7 doxorubicin-resistant breast cancer cells. This is achieved by inhibiting the transcription and translation of complex I structural proteins, which in turn, blocks electrons passage down the ETC to form ROS[50] ”(line 212-216)

These new lncRNAs were also included in Table 1.

The conclusion section has also been modified to better serve the new format of the review.

“The dysregulation of oxidative stress is an essential aspect that allows cancer cells to take the advantage over the neighboring cells. However, ROS are a double-edged sword, as excess oxidative stress induces growth inhibition and cell death. Conversely, cancer cells have developed elaborate accommodating the increased level of ROS through multiple mechanisms. Several lncRNAs regulate the ROS homeostasis at the mitochondrial level, influencing the level of ROS generated in the ETC. Furthermore, by inducing Nrf2 gene expression, acting as ceRNA for Nrf2 mRNA targeting miRNA, preventing Nrf2 protein cytoplasmatic and nuclear inactivation and acting as effectors in the Nrf2-induced antioxidant response, lncRNAs sustain the activity of the main transcription factor of antioxidant response. LncRNAs also sustain the Warburg effect in cancer cells by promoting PKM2 isoform expression, another effective mechanism that perpetuates the dysregulated oxidative balance. Moreover, lncRNAs can act in synergy with ROS in orientating the cancer cell toward proliferation or apoptosis by modulating the expression of numerous important genes in the ROS pathways (e.g., TP53, HIF-1-α,).

The specificity of an unstable redox balance and patterns of lncRNAs expression to cancer cells offer a great therapeutic opportunity in the fight against cancer. However, although some clinical trials targeting lncRNAs have shown encouraging results, the majority of the research is still in the preclinical phase and further investigation and development through in vitro studies and animal models are required before lncRNA-targeted medicine can become a mainstream approach.”

Reviewer 2 Report

Kacso and colleagues describe the relationship between lncRNAs and ROS metabolism in cancer. Furthermore, they highlight the importance of this crosstalk in maintaining the balance of ROS in cancer cells, supporting their proliferation. These aspects make lncRNAs fascinating targets for innovative therapeutic approaches. 

Overall, the topic is interesting and still pioneering. This certainly makes it worthy of discussion. Nevertheless, there are a few points that the authors could improve on to make it more accessible. 

Major:

The introduction is too short and not sufficient to introduce the reader to the topic. The authors should already introduce the concepts of ROS and lncRNAs more extensively at this point. In particular, there is a lack of discussion of the ROS balancing mechanism and its main players in cancer. Such a change would help to emphasise the importance of the topic in the following sections. Similarly, section IV lacks an adequate introduction on the Warbourg effect. 

Section VI, Conclusions and further perspectives, is not sufficiently comprehensive. The authors only marginally refer to possible therapies without discussing in detail where these developments stand (whether in the pre-clinical, clinical phase or still belonging to basic research). 

Overall, although the review deals with an interesting topic, it would benefit from an extensive review of the organisation of the text in order to make it easier to read. 

Minors:

line 73: lncrRNA instead of lncRNA

line75: Table 1 does not have an adequate title.

Lines 159 and 231: Fiugre instead of Figure

Line 373-377: the authors should provide an adequate reference for this statement. 

Author Response

Dear esteemed referee,

Thank you for allowing us to submit a revised draft of the manuscript “Reactive oxygen species and long non-coding RNAs, an unexpected crossroad in cancer cells” to IJMS special issue “Role of ncRNAs Classes as Biomarkers for Diagnostic and Prognosis in Cancer 2021” . We appreciate the time and effort that you have dedicated to providing your valuable feedback on our manuscript. We have been able to incorporate changes to reflect most of the insightful suggestions provided.  The changes are inserted with track changes in the revised manuscript and are also presented below. When refereeing to a line in the revised manuscript, the view with “show insertion and deletions” is used

Major:

  1. The introduction is too short and not sufficient to introduce the reader to the topic. The authors should already introduce the concepts of ROS and lncRNAs more extensively at this point. In particular, there is a lack of discussion of the ROS balancing mechanism and its main players in cancer. Such a change would help to emphasise the importance of the topic in the following sections.

Response: We agree with the reviewer’s assessment. Accordingly, the introduction has been extensively restructured so that it better introduces the reader to the subject of ROS and their importance in cancer cells. The general presentation of lncRNAs and their importance in cancer was also moved to the introduction. We then proceed to discuss in the section “LncRNAs and the generation of mitochondrial ROS in cancer” (previously titled “LncRNAs and ROS in cancer cells. Connecting the dots”), the influence of lncRNAs in the mitochondrial generation of ROS.

  1. Similarly, section IV lacks an adequate introduction on the Warbourg effect. 

Similarly, section IV has been updated to include a more extensive introduction to the Warburg effect:

From “Now generally accepted as a core hallmark of cancer, the metabolic reprograming shifts the use of glucose from an energy producing molecule, to a biomass source, used in the synthesis of essential proteins for the rapidly dividing cancer cells[98]” (line 231-234)

To “Now generally accepted as a core hallmark of cancer, the metabolic reprogramming (Warburg effect) shifts the use of glucose from an energy-producing molecule, to a biomass source, used in the synthesis of essential proteins and nucleic acids for the rapidly dividing cancer cells[115]. This is attained by redirecting a large part of the glucose intake of the cell, away from oxidative phosphorylation and towards the glycolytic pathway. While in normal cells that are not actively growing, this process happens only under hypoxic conditions, in the cancer cell, through activation of oncogenes such as MYC and PI3K/AKT signaling, this happens even in normoxic conditions, hence the term aerobic glycolysis[115] ” (line 314-322)

  1. Section VI, Conclusions and further perspectives is not sufficiently comprehensive. The authors only marginally refer to possible therapies without discussing in detail where these developments stand (whether in the pre-clinical, clinical phase or still belonging to basic research). 

Response: Thank you for pointing this out. Section V has been revised to include a more precise and comprehensive analysis of the current status of treatments targeted at oxidative-stress related lncRNAs.

“In search for novel therapeutic targets, lncRNAs have emerged as valid contenders due to their cancer cell line specificity and low level of expression in other cells. A perfect lncRNA target from a therapeutic standpoint would check 4 essential boxes: predominant expression of a single isoform (insignificant alternative splicing), cancer cell restricted expression, specific function and to have a highly conserved structure throughout species to allow for extrapolation from in vivo studies[156,157]. Although no single lncRNA satisfies all the above conditions, several intriguing targets have been identified. Multiple methods for such targeting have been proposed. Firstly, silencing lncRNA genes transcription using systems such as clustered regularly interspaced short palindromic repeats (CRISPR)/Cas9 or zinc finger nucleases (ZFNs) have been successfully utilized in vitro, resulting in suppression of the targeted lncRNA effect[158,159]. Nonetheless, given that lncRNAs can share promoters with other coding or non-coding genes, using the CRISPR/Cas9 system for targeting lncRNAs comes with the risk of inadvertently deregulate neighboring genes[160]. Secondly, disrupting the interaction between a lncRNA and its specific target would in theory alter the transcription of the genes normally regulated by the interaction. For example, using a newly designed molecule named AC1N0D4Q, Ren et al. managed to block the interaction between lncRNA HOTAIR and the EZH2 catalytic subunit of the PCR2 complex, which resulted in in a reduction of breast cancer cell migration and invasion in both in vitro and animal models[161]. However, controversy still exists over the specificity of some of these lncRNA-gene regulating complexes interactions, given that they function based on scaffolding rather than complementarity – as RNA-DNA interactions do – [162–165]. Thirdly, reducing the level of expression of lncRNAs by degradation of the non-coding transcripts per se is another approach that proved efficient in several studies. This can be achieved by siRNA or antisense oligonucleotides (ASO). Indeed, administration of MALAT1 targeting ASOs reduced metastasis in both human lung cancer xenograft models[58] and in mouse mammary tumor models[166], while siRNA knockdown of MALAT1 reduced cell motility in in vitro models[167]. Moreover, clinical trials using these methods have also rendered positive results. Nanocomplex delivery of MALAT1 targeting siRNA in glioblastoma patients increased the sensitivity of the tumor to temozolomide (TMZ) treatment[168], while a novel ASO drug against non-coding mitochondrial RNA demonstrated good tolerance in a phase one clinical study[169,170].” (line 463-495)

In the conclusions section, we added a statement to better emphasize the road forward: “However, although some clinical trials targeting lncRNAs have shown encouraging results, the majority of the research is still in the preclinical phase and further investigation and development through in vitro studies and animal models are required before lncRNA targeted medicine can become a mainstream approach.” (line 509-514)

The conclusion section has also been modified to better serve the new format of the review.

“The dysregulation of oxidative stress is an essential aspect that allows cancer cell to take the advantage over the neighboring cells. However, ROS are a double-edged sword, as excess oxidative stress induces growth inhibition and cell death. Conversely, cancer cells have developed elaborate accommodating the increased level of ROS through multiple mechanisms. Several lncRNAs regulate the ROS homeostasis at the mitochondrial level, influencing the level of ROS generated in the ETC. Furthermore, by inducing Nrf2 gene expression, acting as ceRNA for Nrf2 mRNA targeting miRNA, preventing Nrf2 protein cytoplasmatic and nuclear inactivation and acting as effectors in the Nrf2 induced antioxidant response, lncRNAs sustain the activity of the main transcription factor of antioxidant response. LncRNAs also sustain the Warburg effect in cancer cells by promoting PKM2 isoform expression, another effective mechanism that perpetuates the dysregulated oxidative balance. Moreover, lncRNAs can act in synergy with ROS in orientating the cancer cell toward proliferation or apoptosis by modulating the expression of numerous important genes in the ROS pathways (e.g., TP53, HIF-1-α,).

The specificity of an unstable redox balance and patterns of lncRNAs expression to cancer cells offer a great therapeutic opportunity in the fight against cancer. However, although some clinical trials targeting lncRNAs have shown encouraging results, the majority of the research is still in the preclinical phase and further investigation and development through in vitro studies and animal models are required before lncRNA-targeted medicine can become a mainstream approach.”

Overall, although the review deals with an interesting topic, it would benefit from an extensive review of the organisation of the text in order to make it easier to read. 

We hope that the new text organization has addressed this issue.

Minors:

- lncrRNA (line 73) was change to lncRNA (line

-an adequate title was added to Table 1 “Important lncRNAs in the regulation of oxidative stress and their impact on cancer cells” (line 144-145)

- Fiugre (lines 159 and 231) was changed to Figure (line 242,323)

Additionally, the manuscript was rechecked for any other spelling or grammatical mistakes.

-Line 373-377: the authors should provide an adequate reference for this statement. 

Here we provided the references from two comprehensive and very well-structured papers. The statement “predominant expression of a single isoform (insignificant alternative splicing)” (line 65-466) derives from the information in the paper entitled “The landscape of long noncoding RNAs in the human transcriptome”. The statement “specific function and to have a highly conserved structure throughout species to allow for extrapolation from in vivo studies” derives from the reading of the paper entitled “ (line 467-469)From biomarkers to therapeutic targets-the promises and perils of long non-coding RNAs in cancer”